# Morbidity and Mortality Due to *Schistosoma mansoni* Related Periportal Fibrosis: Could Early Diagnosis of Varices Improve the Outcome Following Available Treatment Modalities in Sub Saharan Africa? A Scoping Review

**DOI:** 10.3390/tropicalmed5010020

**Published:** 2020-02-03

**Authors:** Daniel W. Gunda, Semvua B. Kilonzo, Paulina M. Manyiri, Robert N. Peck, Humphrey D. Mazigo

**Affiliations:** 1Department of Medicine, Catholic University of Health and Allied Sciences, Mwanza 1464, Tanzania; 2Department of Parasitology, Catholic University of Health and Allied Sciences, Mwanza 1464, Tanzania

**Keywords:** Schistosoma related liver fibrosis, periportal fibrosis, esophageal varices, esophagogastric varices, non-invasive predictors, Sub Saharan Africa, primary prevention, secondary prevention, non-selective B-blockers, endoscopic sclerotherapy, endoscopic variceal ligation

## Abstract

Schistosomiasis affects about 240 million people worldwide and *Schistosoma mansoni* alone affects over 54 million people leaving 400 million at-risk especially in Sub Saharan Africa (SSA). About 20 million people are currently suffering from complications of chronic *S. mansoni* infection and up to 42% of those infected have been found with periportal fibrosis (PPF). About 0.2 million deaths are attributed to chronic *S. mansoni* every year, which is mainly due to varices. Death occurs in up to 29% of those who present late with bleeding varices even with the best available in-hospital care. The diagnosis of varices before incident bleeding could potentially improve the outcome of this subgroup of patients is SSA. However, there is no prior review which has ever evaluated this issue detailing the magnitude and outcome of varices following available treatment modalities among patients with Schistosoma PPF in SSA. This review summarizes the available literature on this matter and exposes potential practical gaps that could be bridged to maximize the long-term outcome of patients with *S. mansoni* related PPF in SSA. A total of 22 studies were included in this review. The average prevalence of varices was 82.1% (SD: 29.6; range: 11.1%–100%) among patients with PPF. Late diagnosis of varices was frequent with average bleeding and mortality of 71.2% (SD: 36.5; range: 4.3%–100.0%) and 13.6% (SD: 9.9; range: 3.5%–29%), respectively. Predictors were reported in seven (31.8%) studies including platelet count to splenic diameter ratio (PSDR) for prediction large varices in one study. Active *S. mansoni* infection was very prevalent, (mean: 69.9%; SD: 24.4; range: 29.2–100.0%). Praziquantel could reverse PPF and use of non-selective B-blockers reduced both rebleeding and mortality. Use of sclerotherapy for secondary prevention of variceal bleeding was associated with high rebleeding and mortality rates. Conclusions: This review shows that varices due to schistosomal PPF are a big problem in SSA. However, patients are often diagnosed late with fatal bleeding varices. No study had reported a clinical tool that could be useful in early diagnosis of patients with varices and no study reported on primary and effective secondary prevention of bleeding and its outcome. Regular screening for *S. mansoni* and the provision of Praziquantel (PZQ) is suggested in this review. More studies are required to bridge these practical gaps in Sub Saharan Africa.

## 1. Background 

Schistosomiasis is a neglected tropical disease (NTD) that causes high morbidity and mortality mostly in African countries. Estimates of the global burden of diseases indicate that schistosomiasis affects more than 240 million people globally and more than 700 million are at risk of infection [1]. The morbidity and mortality due to *Schistosoma mansoni* alone are correspondingly high. In the meantime, *S. mansoni* is reported to affect about 54 million of those who are infected and 400 million more people are at risk of *S. mansoni* infection particularly in Sub Saharan Africa (SSA) [1,2]. Sub Saharan Africa is the most affected part of the world where over 90% of the global burden of schistosomiasis is concentrated leading to an estimated 0.2 million loss of lives annually mainly due to chronic *S. mansoni* infection [1,3].

A combined preventive strategy including access to safe water, improved sanitation, hygiene education, snail control, and targeted preventive chemotherapy mass drug administration is the most effective way for elimination and control of schistosomiasis transmission [4]. Sub Saharan Africa represents a region in the world where preventive measures against transmission of schistosomiasis have not worked. Tanzania is the most affected country in this region after Nigeria. In Tanzania, for instance, high transmission of more 50% still occurs among communities that are engaged in fresh water-related activities [1,2] with school-aged children, women, and fishermen being at the highest risk of infection. Use of Praziquantel (PZQ) in mass drug administration (MDA) is common in most SSA countries mainly targeting the school-aged children however there is a rapid and high re-infection rates post PZQ preventive treatment [5,6,7]. Additionally, occupational activities, poor sanitation, and lack of access to safe water significantly compromise the mitigation of *S. mansoni* transmission and most people are recurrently infected.

Recurrent *S. mansoni* infection frequently induces Schistosoma related hepatosplenic disease which is usually marked by the formation of hepatic granulomas presenting clinically with hepato splenomegaly. In the later stage of the disease, granulomas progressively transform into fibrous tissues in periportal areas [8]. Community-based studies suggest that periportal fibrosis (PPF) is the commonest complication of chronic *S. mansoni* infection and non-cirrhotic liver fibrosis-related portal hypertension. In Tanzania, periportal fibrosis is reported between 22% and 42% of studied patients [9,10] and the mortality from this complication is exceptionally high. According to WHO reports about 0.2 million lives are lost every year in SSA due to this complication [1] and in some settings, the mortality following bleeding esophageal varices has been reported between 25% and 29% even with the best available in-hospital care [11,12].

In Tanzania, up to 42% of *Schistosoma mansoni* infected patients studied sonographically were found to have periportal fibrosis [10,13]. In the advanced stage, PPF is commonly attended by portal hypertension and formation of esophagogastric varices which may potentially bleed with high mortality. According to WHO, an estimated 0.2 million deaths occur every year in SSA due to this complication [1]. The mortality following bleeding varices is high which maybe more than 25.0% even with the best in-hospital care [11]. Our question of interest is whether an early diagnosis of esophageal varices before incident bleeding and institution of preventive treatment against bleeding can potentially reduce the bleeding and the associated mortality among patients with *Schistosoma mansoni* related hepatic fibrosis.

Endoscopic screening of all patients with PPF would potentially diagnose most of the attendant varices before bleeding similar to what is done among those patients with liver cirrhosis related portal hypertension [14]. We believe that early detection of varices and prevention of bleeding using B-blockers and band ligations will potentially prevent bleeding subsequently reducing mortality among patients with periportal fibrosis similar to cirrhosis related varices [15]. However, there is no prior review which has ever evaluated this issue detailing the magnitude and outcome of esophageal varices following available treatment modalities among patients with *S. mansoni* related PPF in SSA. This review is an attempt to summarize the available literature regarding this subject and expound potential practical gaps that could be bridged to maximize the long-term outcome of patients with *S. mansoni* related PPF in SSA.

## 2. Methods 

This review is based on a systematic search of Pub Med, Google Scholar, and Web of science and citation lists of relevant publications according to the PRISMA checklist [16].To obtain the relevant kinds of literature from these electronic databases the following keywords were used: esophageal varices or esophagogastric varices and Schistosoma or *Schistosoma mansoni* or schistosomal periportal fibrosis and Sub Saharan Africa or Africa or “specific country names”. The search considered all publications in English with original data on esophageal varices among patients with chronic Schistosoma liver disease in Sub Saharan Africa who were treated non-surgically. The database search and summary of articles were independently done by two authors. The abstracts were screened for relevance and the eligible articles were retrieved for review (Figure 1). The extracted data were recorded in an excel sheet for analysis and it was summarized in a table for synthesis (Table 1).

## 3. Search Results 

A total of 22 studies were included in this scoping review as summarized in Figure 1. Most studies, 13 (59.1%), were prospective in design with a mean sample size of 112 participants (SD: 96.8; range: 32–492). All studies had upper gastrointestinal endoscopy done in their participants and most, 20 (90.1%), had available ultrasound reports. About a third, six (27.3%), of the studies, included participants with periportal fibrosis who subsequently underwent endoscopic screening for esophageal varices and eight(36.4%) studies included participants with esophageal varices who subsequently underwent abdominal ultrasound scanning for the cause of portal hypertension. The rest, eight (36.4%) involved patients who either presented with upper gastrointestinal bleeding (UGIB) or *S. mansoni* infection and they were investigated for the presence of varices among others. Several other tests were reported in some studies including hepatitis B and C test, liver biopsy, complete blood count (CBC), serum albumin, alanine aminotransferase (ALT), aspartate aminotransferase (AST), and test for active *S. mansoni* infection.

By ultrasound, PPF was reported in 18 (81.8%), studies and esophageal varices in all 22 (100%) studies including 14 (63.6%) studies that enrolled patients who were known to have varices. The average prevalence of varices was 82.1% (SD: 29.6; range: 11.1%–100%). A report on bleeding esophageal varices was available in 18 (81.8%) studies inclusive of seven (31.8%) studies that studied patients who presented with UGIB with average bleeding of70.4% (SD: 35.2; range: 4.3%–100%). Mortality was reported in seven (31.8%) studies with an average mortality of 13.6% (SD: 9.9; range: 3.5%–29%) following bleeding events. 

Predictors were reported in eight (36.4%) studies including predictors of esophageal varices and bleeding events. One study reported on a potential tool to assist in the diagnosis of bleeding varices due to *Schistosoma mansoni* and another one reported on a potential role of platelet count to splenic diameter ratio (PSDR) in predicting large varices among those diagnosed with esophageal varices.

A total of eight (36.4%) studies, reported doing tests for active *S. mansoni* either by circulating cathodic antigen (CCA) or Kato Katz (KK) methods with a mean prevalence of 69.9% (SD: 24.4; range: 29.2–100.0%). The use of praziquantel (PZQ) following a positive *S. mansoni* test was reported in two (9.1%) studies and it was demonstrated in one study that PZQ could reverse *S. mansoni* related hepatosplenic disease. No report was available on primary prevention of variceal bleeding in SSA. Four (18.2%) studies reported on the use of non-selective B-blockers (NSBB) for secondary prevention of variceal bleeding and seven (31.8%) studies reported on the use of sclerotherapy (SCL) for secondary prevention of variceal bleeding however with high rebleeding rate of up 32.0% and a high mortality reaching 29% (Table 1).

## 4. Discussion

The main objective of this review was to summarize the available literature with original data on the magnitude, outcome of esophageal varices and elucidate on potential practical gaps following available treatment modalities among patients with *Schistosoma mansoni* related periportal fibrosis in SSA. In this review, we found that esophageal varices are commonly encountered in patients with periportal fibrosis. Most studies had reported a high proportion of late diagnosis of esophageal varices at the bleeding stage with significantly high mortality. The presence of varices correlated with advanced fibrosis and the male gender. Splenomegaly and source of water had high sensitivity in predicting bleeding from varices due to *S. mansoni* in one study. The PSDR was studied in one study for prediction of large varices among those with varices. No tool was reported for the prediction of varices before incident bleeding. Active *S. mansoni* infection was commonly reported and the use of PZQ reversed hepatic fibrosis. Use of NSBB reduced rebleeding and mortality while sclerotherapy was associated with high failure rates and death.

A wide range of prevalence of esophageal varices among patients with Schistosoma related PPF was observed in this review. Apart from the studies that included patients who were known to have varices, the highest prevalence was 80% that was reported in a recent study by Opio et al. from Uganda among 107 participants who had *S. mansoni* related PPF [17]. The lowest prevalence of varices of 12.1% was reported much earlier in 1997 by Madwar et al. from Egypt among 120 participants [18]. The small rate of esophageal varices in Egypt could be due to reduced transmission of *S. mansoni* following the construction of Aswan dam and effective control programs [37]. In areas where transmission has remained high, most people are recurrently infected with a high risk of severe PPF. In this review, these patients were more likely to have varices that were even more likely to present with bleeding [17,19].

The available body of literature indicates that the diagnosis of esophageal was often made late already in a bleeding state with unacceptably high mortality. Apart from studies which enrolled patients with bleeding varices alone, bleeding was reported in 16 (72.7%) studies and the prevalence was between 4.3% and 81.0%. The highest bleeding rate of 81.0%was reported by Davidson et al. among 62 patients [20]. Though the lowest bleeding event was 4.3% as reported by Saad et al. [21], most studies reported high prevalence rates of more than 45% including those from Zambia, Uganda, and Kenya [17,22,23].

Studies that enrolled patients with upper gastrointestinal bleeding emphasize furthermore on the fact that *S. mansoni* related varices are a common cause of bleeding in SSA which usually ends up with the fatal outcome even with the best in-hospital treatment. For instance, Awilly et al. studying patients who were admitted following hematemesis indeed found out that 70%of their participants had bleeding varices where active *S. mansoni* infection was a very common encounter in up to 60% of the studied participants [24]. The WHO estimates that due to chronic *S. mansoni* infection about 0.2 million people die every year in SSA [38]. Among 91 patients who underwent endoscopy in a report by Awilly due to vomiting blood 13 (14.2%) were reported to die within 2 months of follow up [24]. Studies from Sudan with the use of sclerotherapy for secondary prevention of bleeding reported a high rebleeding rate of up to 43.3% with even higher mortality rates of between 25% and 29% [11,12]. These mortality rates are unacceptably high thus strongly calling for deliberate measures to improve the outcome of this subgroup of patients in SSA.

Endoscopic screening of all patients for varices at the time of diagnosis of PPF can potentially diagnose attendant varices the earliest before bleeding as demonstrated in a study by Saad et al. In this study patients with schistosomiasis were sonographically screened for *Schistosoma mansoni* related liver fibrosis and those who were found to have PPF were subsequently screened for esophageal varices. Among 95 participants who had PPF, 67% of them were found to have varices and only 4.3% had bleeding events [21]. This suggests that with well-timed endoscopy most patients with varices can be diagnosed before bleeding and benefit from primary prevention of bleeding and further reduce mortality.

Due to resource restriction and a high burden of *Schistosoma mansoni* in SSA, clinicians would need a set of independent predictors and non-invasive clinical tools to assist in the selection of patients at high risk of having varices for prioritized endoscopic intervention. Several independent predictors for varices or bleeding events were reported in studies incorporated in this review, including male gender, advanced liver fibrosis [21,22,24], reduced platelet count, an older age than 40years, female gender, and hepatitis C co-infection [17,18]. Other factors including water source, positive Schistosoma test, ascites, large portal vein, and massive splenomegaly were reported to predict bleeding due to *S. mansoni* related varices [24]. These predictors are all relevant and scientifically logical except for the gender that may be due to the local dominance of a particular gender as suggested in studies from Tanzania and Uganda [17,24].

A combination of these factors with other predictors could potentially come out with a clinically sensitive non-invasive tool. In a study by Chofle et al., a combination of a source of water and dilated portal vein were found to have a higher performance in predicting esophageal varices as a cause of hematemesis [24]. Platelet count splenic diameter ratio (PCSDR) is another tool that has been validated in multiple studies in picking esophageal varices among patients with cirrhosis related portal hypertension. Two studies elsewhere had reported this tool being sensitive enough in sorting outpatients at high risk of having esophageal varices among patients with PPF including a study from Saudi by Adnan [39] and Xu XD et al. from China [40]. In SSA this tool was used in one study among patients with varices for prediction of large varices [25]. The utility of this tool in the primary prediction of varices has not been assessed in SSA according to the available body of literature among patients with Schistosoma PPF.

Based on the current review, it is evident that active *S. mansoni* infection is very frequent among patients with periportal fibrosis in SSA, suggesting that even at this stage of complications patients with PPF should routinely be investigated for active *S. mansoni* infection regularly provided with PZQ since also mass drug administration is only restricted to the school-aged population in most countries [4]. The available evidence indicates that PZQ reduces infection intensity and significantly reverses granuloma and liver fibrosis with its attendant complications, thus PZQ could be used for pre-primary prevention [26,41,42]. 

Non-selective B-blockers among PPF patients with varices have been reported elsewhere as a potential pharmacological measure for primary and secondary prevention of variceal bleeding [43]. The NSBB could be combined with variceal ligation (EVL) among those with large varices either with or without prior bleeding [44,45,46]. In the current body of literature in SSA, no report was available on the use and effect of NSBB on primary prevention of variceal bleeding. Otherwise, NSBB was shown to reduce both rebleeding rate and mortality among those with bleeding varices [27,28]. No study was available on the use of EVL for primary or secondary prevention of variceal bleeding in this review in SSA. Sclerotherapy was reported as a secondary preventive measure for rebleeding [28,29,30,31] however was attended by high rates of complications and rebleed rates of up to43%and subsequently high mortality of up 29% [11,12,31]. A combination of SCL had rapid eradication of varices with high recurrence rate when NSBB was discontinued [32]. In Tanzania EVL and NSBB are usually combined for both primary and secondary prevention of bleeding varices in patients with PPF, with a pending evaluation of its long-term effect [47].

## 5. Conclusions 

The current review has found out that esophageal varices are a very common encounter among patients with *S. mansoni* related PPF in SSA. However, most patients tend to be diagnosed late with significantly high prevalence of bleeding and unacceptably high mortality. There is still limited data regarding potential non-invasive tools that are sensitive enough in predicting patients at high risk of having esophageal varices before incident bleeding for prioritized endoscopic intervention. We are still also lacking data regarding any effective primary and secondary prevention of variceal bleeding and its long-term outcome in this region. Routine screening for *S. mansoni* and provision of PZQ among patients with PPF is highly suggested in this review. More studies are required to bridge these gaps to improve the outcome of this subgroup of patients. We are currently conducting a study to address some of these practical gaps including assessment of a potential tool for primary prediction of esophageal varices and prospectively assessing the outcome of patients with *S. mansoni* related liver fibrosis and its attendant varices following the use of praziquantel, non-selective B-blockers, and endoscopic variceal ligation.

## Figures and Tables

**Figure 1 tropicalmed-05-00020-f001:**
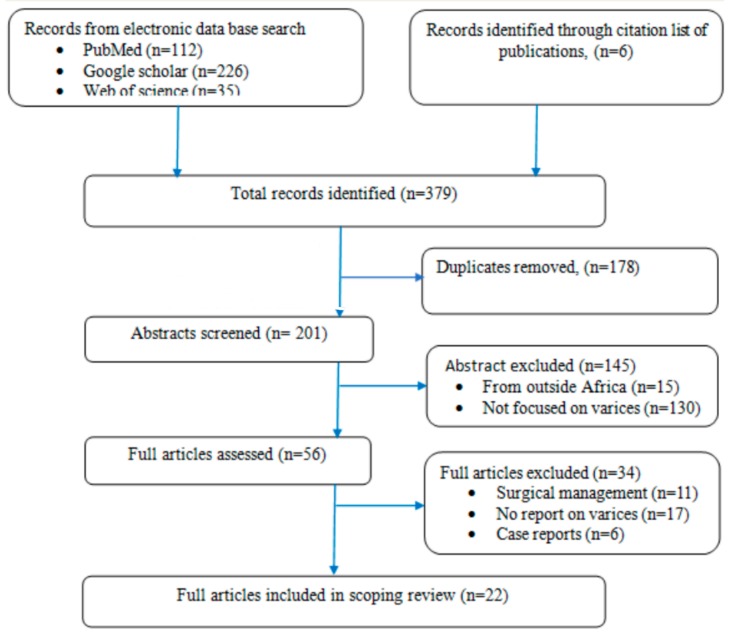
Flow chart indicating selection of articles included in this review.

**Table 1 tropicalmed-05-00020-t001:** Study objectives, design, outcomes, and predictors among 22 studies included in the review

Author, Country	Study Description	Assessment/Treatment	Important Findings and Outcome
**Bessa S.M. (1985)**, **Sudan [11]**	**Prospective study**; 60 patients with PPF and bleeding varices underwent SCL and were followed up for 18.4 months	**Investigations**: ultrasound, stool for *S. mansoni* ova and, upper GIT endoscopy **Treatment**: sclerotherapy	Controlled acute bleeding in 17 (85%; *n* = 20) patients 11 (18.3%) patients rebleed 10(25%) patients died
**El-Zayadi A. (1988), Sudan [12]**	**Prospective (RCT) study**; 118 patients with PPF and bleeding varices were randomized to SCL and medical resuscitation, and all were followed up for 21 months	**Investigations**: ultrasound, stool for ova, endoscopy **Treatment**: SCL, (63 patients); resuscitation with BT, vasopressin,(55 participants)	9 (14.3%)died in SCL group, 16 (29%) died in the medical group Rebleeding was statically similar in the two groups
**Opio CK. (2016); Uganda [17]**	**Cross-sectional study**; 107 participants with *S. mansoni* liver fibrosis and esophageal varices, were assessed to determine lifetime bleeding and its predictors.	**Investigations**: ultrasound, upper GIT endoscopy, FBP, HBVsAg, HCVAb, malaria test, urine CCA, stool for ova **Treatment**: not specified	61 (57%) participants had bleeding at least two times Predictors of bleeding were being female, age>40years, advanced liver fibrosis
**Madwar MA. (1997); Egypt [18]**	**Prospective study**;120 liver disease patients, were followed up for predictors of bleeding incident bleeding; were followed up for 18 ± 7.3 months	**Investigations**: ultrasound, Hepatitis B and C test (HBsAg, HCV-ab), complete blood count, upper GIT endoscopy, **Treatment**: not specified	12.1% with PPF had bleeding Bleeding predictors: cherry-red spots, larger varices, gastric varices and prothrombin time
**Ritcher J. (1992); Sudan [19]**	**Cross-sectional study**; correlating UTS findings of Schistosoma liver disease to clinical presentation. A total of of 32 patients with PPF were included	**Investigations**: ultrasound, upper GIT endoscopy, **Treatment**: Praziquantel	59%(*n* = 32) of participants bled Predictors of bleeding: large portal vein diameter and advanced liver fibrosis
**Davidson R.N. (1991); Zimbabwe [20]**	**Cross-sectional study**; 62 patients with varices were assessed for periportal fibrosis (PPF) and *S. mansoni* infection.	**Investigations**: upper GIT endoscopy, ultrasound, rectal snip for *S. mansoni*. **Treatment**: not specified	46 (74%) patients had PPF 50 (80.1%) had bleeding 19 (98%, *n* = 20) snips were positive for ova in of PPF
**Saad AM. (1991); Sudan [21]**	**Cross-sectional study**; screening for PPF and varices in *S. mansoni* endemic area. Randomly included 20% of the villagers (660 participants) from two villages.	**Investigations**: stool for *S. mansoni* ova, ultrasound, upper GIT endoscopy in 76 participants with PPF **Treatment**: not specified	325(29.2%) had *S. mansoni* 95 (14.4%) had PPF by UTS 45 (59.2%, *n*=76) had varices Three (4.0%, *n* = 76) had bleeding esophageal varices
**Payne L. (2013); Zambia [22]**	**Cross-sectional study**; 110 patients with PPF underwent an ELISA test for active *S. mansoni* and upper GIT bleeding symptom assessment	**Investigations**: ELIZA for active *S. mansoni* **Treatment**: not specified	By ELISA 97(88%) had active *Schistosoma mansoni* infection 63 (53%)reported having upper GIT bleeding incidences
**De Cock KM. (1982); Kenya [23]**	**Cross-sectional study**; 68 patients with portal hypertension and proved esophageal varices were evaluated for the cause of portal hypertension, bleeding, and associated *S. mansoni*	**Investigations**: ultrasound, liver biopsy, stool for *S. mansoni*, rectal snip, upper GIT endoscopy **Treatment**: not specified	29.4% of participants were found to have Schistosoma PPF Variceal bleeding in 50% PPF correlated very well to *S. mansoni*
**Chofle AA. (2014); Tanzania [24]**	**Prospective study**; 124 patients admitted for UGIB. Etiology and predictors of bleeding, active, *Schistosoma mansoni* infection. A two month follow up was done and mortality was reported	**Investigations**: ultrasound, FBP, urine CCA, ALT, AST, PTT, PT, HCVAb, HBsAg, Albumin, BRBN **Treatment**: transfusion, other details not reported	70% of patients had varices 60% had active *S. mansoni* 13/91(14.3%) died within 2 months of follow up PVD and water source predicted bleeding varices
**Hassan M.A. (2018); Sudan [25]**	**Cross-sectional study**; 100 with PPF and esophageal varices were studied for non-invasive predictors of large varices	**Investigations**: ultrasound endoscopy, FBP, Hepatitis B, and C test, **Treatment**: not specified	81 (81.0%) had large varices 41 (41.0%) had bleeding Low PLT and PPF G3/4 predicted large varices
**Kheir MM, (2000) Sudan [26]**	**A prospective study**; a pre- and post-MDA survey. The effect of single-dose PZQ was then assessed on Schistosoma related morbidity and mortality.	**Investigations**: stool for *S. mansoni* ova (Kato Katz), ultrasound, endoscopy **Treatment**: Praziquantel single dose (40mg/kg)	PZQ single dose reduced Infection from 53% to 43% PPF from 14% to 10% Varices from 30% to 47%
**El Tourabi H. (1994); Sudan [27]**	**Prospective (RCT) study**; 82 patients with esophageal varices were randomized to Propranolol and placebo for 24 months	**Investigations**: ultrasound, upper GIT endoscopy **Treatment**: Propranolol and placebo	Use of Propranolol led to a significant reduction of rates of rebleeding and mortality (*p*<0.02) as compared to placebo
**Dowidar N. (2002); Egypt [28]**	**Prospective (RCT) study**; 40 patients with bleeding varices were injected with 2.5% or 5% ethanolamine. A 2 year follow up was done for rebleeding	**Investigations**: ultrasound, upper GIT endoscopy **Treatment**: SCL 2.5%/and SCL 5% for prevention of variceal rebleeding	SCL with 5% has rapid eradication of varices with similar rebleeding rates over the 2 years of follow up.
**Mostafa I. (1997); Egypt [29]**	**Prospective study**; 100 patients with bleeding gastric varices underwent SCL for secondary prevention of bleeding	**Investigations**: upper GIT endoscopy, ultrasound **Treatment**: sclerotherapy	12.5% rebled in 24 h 6.25% of patients died
**Maurizio R. (2000); Uganda [30]**	**Prospective study**; 34 patients with PPF and bleeding varices underwent sclerotherapy and followed up for 10 ± 2.1 months	**Investigations**: endoscopy, stool for Schistosoma ova, ultrasound **Treatment**: sclerotherapy	Varices eradication achieved in 28 (82.3%) patients Rebleeding, four(11.8%, *n* = 34) 2 (5.9%, *n* = 34) patients died
**Mundawi HM. (2007); Sudan [31]**	**Prospective study**; 118 patients with PPF and bleeding varices. Underwent SCL and were followed up for rebleeding incidences	**Investigations**: ultrasound, upper GIT endoscopy **Treatment**: SCL to prevent secondary variceal bleeding	32% participants rebled Mortality was 3.5% Rebleeding was predicted by advanced liver fibrosis
**Dowidar N. (2005); Egypt [32]**	**Prospective (RCT) study**; 40 patients with PPF and bleeding varices, Randomized to SCL and NSBB + SCL for secondary eradication of varices	**Investigations**: ultrasound, endoscopyTreatment: SCL alone vs. SCL and Propranolol 40mg od	In SCL/NSBB group fewer SCL sessions were required to eradicate varices NSBB discontinuation caused a variceal recurrence
**De-Cock M. (1983); Kenya [33]**	**Cross-sectional study**; 85 portal hypertension patients and varices. Assessed for cause of portal hypertension and bleeding	**Investigations**: ultrasound, stool for *S. mansoni* ova, hepatitis B test, endoscopy **Treatment**: not specified	25 (29.4%) found with PPF 18 (72%) had *S. mansoni* Of those 52% bled Predictors: not reported
**Mudawi H. (2008); Sudan [34]**	**Prospective study**; 143 patients with PPF referred for endoscopy and were assessed for gastric varices and bleeding incidences	**Investigation**: ultrasound, upper GIT endoscopy **Treatment**: not specified	All had esophageal varices Gastric varices; 24 (16.8%) 21.7% PHTN gastropathy Gastric varices rarely bled
**Ravera M. (1996); Uganda [35]**	**Prospective study**; 122 patients with *S. mansoni* were assessed prospectively for development PPF and varices	**Investigations**: stool for *S. mansoni* ova, ultrasound, upper GIT endoscopy **Treatment**: Not specified	45% developed varices Presence of esophageal varices correlated well with the severity of liver fibrosis
**Houston S. (1993); Zimbabwe [36]**	**Cross-sectional study**; 492 participants from *S. mansoni* endemic area were studied for PPF, active infection, and varices	**Investigations**: stool Kato Katz for *S. mansoni* ova, ultrasound, and endoscopy **Treatment**: Praziquantel	PPF found in 129 (26.4%) 59%,(*n*=469) had *S. mansoni* 11.1% (*n*= 18) had varices Splenogally correlated fibrosis

ALT: Alanine aminotransferase; AST: aspartate aminotransferase; CCA: circulating cathodic antigen; BRB: bilirubin; BT: blood transfusion; HBsAg: hepatitis B virus surface antigen; HCVAb: hepatitis C virus antibodies; OGD: oesophagogastroduodenoscopy; MDA: mass drug administration; PPF: periportal fibrosis: PVD: portal vein diameter; PZQ: Praziquantel; UGIT: upper gastrointestinal tract; PLT: platelet; RCT: randomized clinical trial; SCL: sclerotherapy.

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
