# Peer review of "Morbidity and Mortality Due to Schistosoma mansoni Related Periportal Fibrosis: Could Early Diagnosis of Varices Improve the Outcome Following Available Treatment Modalities in Sub Saharan Africa? A Scoping Review"

_tropicalmed, 2020, doi:10.3390/tropicalmed5010020_

Round 1

Reviewer 1 Report

The authors state in line “The primary diagnosis of varices could potentially improve the outcome of this subgroup of patients is SSA though the resource constraint on top of a high burden of S. mansoni in SSA; may lead to inadequate care and unfavorable outcome of this subgroup of patients. “

It is not clear what subgroup of patients the authors are discussing. Patients with PPF? All patients with schisto mansoni? Or patients with PPF and splenomegaly. When they say the primary diagnosis of varices could improve outcome….. They need to state what their hypothesis is… early banding- beta blockers? What are they thinking. This is the crux of the paper and it is not clear to the reader.

Line 65 Schistosoma mansoniinfection and as commonest cause of non-cirrhotic liver fibrosis and portal hypertension. In Tanzania, periportal fibrosis is reported between 42 and 81% of studied patients. 

I pulled the two references. Reference number 6 does not find 81% of individuals with peri-portal fibrosis on scanning. This is an article from 1997 and should be re-reviewed. 

Line 64 – The authors state “Fibro scan studies among Schistosoma mansoni infected patients in community-based studes suggest that periportal fibrosis is the commonest complication of chronic Schistosoma mansoni infection …. References 5,6 are ultrasound studies. There are no studies referenced here on fibro-scanning. This needs to be updated. 

Line 69…mortality I as high as 29% of participants probably because patients are diagnosed late already with bleeding varices with corresponding high mortality. 

The reference number 7 when pulleds ixty patients with esophageal varices caused by schistosomal hepatic fibrosis were managed by injection sclerotherapy. Bleeding was controlled in 17 of 20 patients (85%) with acute bleeding from varices. The remaining 40 patients were either unfit for operation or had a recurrence of variceal bleeding after operation. In a period of follow-up that ranged from 14 to 31 months (mean 18.4 months), 11 patients (18.3%) had a recurrence of bleeding that was controlled and four patients (6.7%) had a fatal recurrence of bleeding. The overall mortality rate was 25% (15 patients), and 34 patients (56.7%) had no recurrence of bleeding

72  Authors state, “The interesting question would be whether the primary diagnosis and management of varices would improve the outcome of this subgroup of patients.”

It is not clear what the authors mean – whether the primary diagnosis and management of varices….. do they mean if individuals are diagnosed earlier with earlier aggressive management would outcomes improve? What management are they referring to? They should be specific. What subgroup of patients are the authors referring to – those with PPF at all? This needs to be clarified as stated above. 

Line 73 Endoscopic screening of all patients with PPF would potentially diagnose most of the patients with attendant varices the earliest similar to what is done among those patients with liver cirrhosis related to portal hypertension. The resource limitations in SSSA and the high burden of S. mansoni morbidity in combination is likely to compromise the delivery of whichever best available care and thus negatively affect the outcome for this subgroup of patients. 

This is not really clear what the authors mean. Also – would you stratify who received endoscopic screening… those with reversal of flow on US or those with ultrasound? Or is that a gap?

It is not clear what the authors are looking at in this article until line 129 The main objective of this scoping review was to summarize the available literature with original data on the magnitude, outcome of esophageal varices and elucidate on potential practical gaps following available treatment modalities among patients with S. mansoni related periportal fibrosis in SSA.  It is not that clear in the introduction. 

Table 1 is very difficult to understand. The types of patients included in the study  should be outlined. How patients were assessed (endoscopy), what type of disease (how many with PPF, etc) and outcomes. Did they receive treatment for schisto? What interventions did they undergo (sclerotherapy/meds) and outcome.  The way it is written now is very difficult to unpack.

I pulled some of the papers. For example, Madwar MA (11) is a prospective study to look at prediction of the variceal haemorrhage in schistosomal and non schistosomal liver disease. In the table the authors refer to those with both as “mixed” type of disease, but this is not explained in the footnote. Since this paper is looking at schistosomiasis alone these individuals need to be pulled out. Also, this paper examines risk factors for bleeding using endoscopy. This was not underscored in the table. 

The paper should be more focused on what is in the literature now on patients with PPF and varices in terms of interventions and outcomes. The table should reflect that and only patiens with PPF and schisto should be looked at – mixed disease should be excluded – it is not part of the goals of the paper. 

The authors should then discuss gaps and suggest areas for research/intervention that make sense in resource poor regions. 

Reviewer 2 Report

Schistosomiasis infection affects about many million people worldwide and according available date data over
700million are at risk of infection. Sub
Saharan Africa (SSA) is the most affected regions.
The current review demonstrates that esophageal varices are a big problem among patients with Schistosomal hepatic fibrosis in SSA. However; patients with varices tend to be
diagnosed late with high rates of variceal bleeding and associated mortality.

According authors no study had reported on a clinical tool to assist in early diagnosis of patients with varices and no study reported on
 primary and effective secondary prophylaxis and its outcome.

The data presented by the authors are a very important warning in terms of public health and derives from a well-conducted study.

However, I believe that this study misses an evaluation of articles that evaluate preventive measures and whether they are a successful story or not in SSA. In the absence of early diagnosis and effective treatment prevention becomes crucial.

For this it is necessary to understand in terms of literature, when the liver changes occur in terms of age, who are the most affected will be men or women? Is it possible to act with school-age children (treatment of infection with the use of praziquantel (PZQ))?
I believe these epidemiological aspects would improve the article and make it more complete.

Round 2

Reviewer 1 Report

The changes are adequate. 

Reviewer 2 Report

The article is now ok. Accepted.